# The *Arabidopsis thaliana–Streptomyces* Interaction Is Controlled by the Metabolic Status of the Holobiont

**DOI:** 10.3390/ijms232112952

**Published:** 2022-10-26

**Authors:** Stéfanie Graindorge, Claire Villette, Sandrine Koechler, Chloé Groh, Sophie Comtet-Marre, Pierre Mercier, Romaric Magerand, Pierre Peyret, Dimitri Heintz, Hubert Schaller, Florence Arsène-Ploetze

**Affiliations:** 1Institut de Biologie Moléculaire des Plantes, CNRS, Université de Strasbourg, 67084 Strasbourg, France; 2Université Clermont Auvergne, INRAE, MEDIS, 63001 Clermont-Ferrand, France

**Keywords:** holobiont, microbiota, plant–bacteria interactions, metabolomics

## Abstract

How specific interactions between plant and pathogenic, commensal, or mutualistic microorganisms are mediated and how bacteria are selected by a plant are important questions to address. Here, an *Arabidopsis thaliana* mutant called *chs5* partially deficient in the biogenesis of isoprenoid precursors was shown to extend its metabolic remodeling to phenylpropanoids and lipids in addition to carotenoids, chlorophylls, and terpenoids. Such a metabolic profile was concomitant to increased colonization of the phyllosphere by the pathogenic strain *Pseudomonas syringae* pv. tomato DC3000. A thorough microbiome analysis by 16S sequencing revealed that *Streptomyces* had a reduced colonization potential in *chs5*. This study revealed that the bacteria–*Arabidopsis* interaction implies molecular processes impaired in the *chs5* mutant. Interestingly, our results revealed that the metabolic status of *A. thaliana* was crucial for the specific recruitment of *Streptomyces* into the microbiota. More generally, this study highlights specific as well as complex molecular interactions that shape the plant microbiota.

## 1. Introduction

Plants host different microorganisms, which interact with the host and each other in a sophisticated functional network [1,2]. Plant-associated microorganisms or microbiota together with the plant constitute a meta-organism called a holobiont [3,4,5,6,7]. Bacterial communities are prominent components of such holobionts. A comprehensive inventory of the bacterial communities interacting with *Arabidopsis thaliana* has been conducted using environmental microbiology and genomics, and many bacteria associated with this model species have been isolated [8,9,10,11]. Bacterial communities are found in the soil near the roots (the rhizosphere), on the surface or within the roots, or in the phyllosphere [4,5,12,13,14]. Several studies have demonstrated a positive impact of those bacteria on plants in numerous ways, by promoting growth, nutrient acquisition, defense against abiotic stresses, and prevention of plant colonization by pathogens [4,5,7].

Some members of the microbiota may be specific to one or few plant genotypes, while others forming the core microbiota are found ubiquitously in plants [8,11]. *Actinobacteria* are ubiquitous in the plant microbiota and among this phylum, *Streptomyces* is the most abundant taxon [4,8,10,11]. Several bacteria affiliated to the *Streptomyces* genus have been recently demonstrated to play a role in plant growth-promoting (PGP) and biocontrol, in part because these bacteria are able to produce antibiotics [15,16,17,18]. In *A. thaliana* but also in *Aconitum carmichaelii*, bacteria belonging to the *Streptomyces* genus protect against biotic stresses [19,20]. This protection is linked to their ability to produce antimicrobial compounds [17,21]. *Streptomyces* were shown to display antagonisms against fungal or bacterial phytopathogens [22,23,24,25], or were found in disease-suppressive soils [26]. A prominent example is that of *Streptomyces* sp. EN27, which induces defense pathways in *A. thaliana* [27], whereas the rhizospheric *Streptomyces* sp. MR14 possesses both antifungal and PGPR capacities in tomato plants [28].

How plants distinguish pathogenic, commensal, or mutualistic bacteria in microbiota is an important question to address [4,29,30]. Several studies have demonstrated that the plant immune system, hormones, and specific metabolites such as flavonoids favor the selection of particular microorganisms [31,32,33,34,35]. Besides flavonoids, isoprenoids are a tremendously important class of compounds involved in essential biological processes such as cell division and elongation (the case of sterols, brassinosteroids, and gibberellins), photosynthesis (chlorophylls and carotenoids), membrane dynamics (sterols), respiration (ubiquinones), or stress response (abscisic acid) [36,37]. Isoprenoid compounds have been given a role in plant–bacteria interactions: linalool emitted by flowers slows the growth rate of bacteria isolated from the *P. digitalis* phyllosphere [38], variations in the cellular quantities of squalene and sterols increase the proliferation of pathogens in *Nicotiana benthamiana* or *A. thaliana* [39], and changes in tricyclic triterpenes derivatives in *A. thaliana* roots had a strong effect on the proliferation of root microbiota [40].

Plant isoprenoid biogenesis occurs in cytosolic, peroxisomal, and plastidial compartments. This starts from the initial production of C_5_ building blocks, namely, isopentenyl diphosphate (IPP) and dimethylallyl diphosphate (DMAPP), by two pathways: the mevalonate pathway in the cytosol and the methylerythritol phosphate (MEP) pathway in the plastids [41]. Loss-of-function alleles of key genes involved in these cytosolic or plastidial pathways result in developmental defects and severe morphogenetic inhibitions. These defects are associated with altered chemical profiles and variations in major metabolic constituents [36]. The enzyme DXS1 (DeoxyXylulose phosphate Synthase 1) is implied in the MEP pathway. A weak allele of the gene encoding, this enzyme was previously characterized in an *A. thaliana* mutant, namely *chs5* (*chilling sensitive* 5; [42]), isolated in a genetic screen for chilling sensitive mutants. This hypomorphic mutant *chs5* displays moderate phenotypic alterations at 22 °C and a chlorotic phenotype when the temperature is shifted to a lower temperature (15 °C). It has been previously demonstrated [43] that the *chs5* mutant had a single base substitution in the *CLA1* gene encoding the DXS enzyme, which changes the aspartate residue to an asparagine residue. This missense mutation D627N was proposed to cause the strong phenotype at low temperature (15 °C) [43]. More recently, Wright and colleagues observed in this mutant a modulation in the content of MEP pathway biosynthetic intermediates in standard *Arabidopsis* growth conditions [44]. 

The aim of the work presented here is to assess how a variation in a metabolic phenotype, such as for instance the isoprenoid composition of a plant, would impact the establishment of the *Arabidopsis* core or specific microbiota. We further characterized the *chs5* mutant using a non-targeted metabolomic approach and observed changes in several groups of metabolites in addition to plastidial isoprenoids, of which some are known to play a role in plant–bacteria interactions. We then challenged the capacity of *chs5* to tolerate a *Pseudomonas syringae* pv. *tomato* strain DC3000 (Pto DC3000) infection [45,46], to assess any change in the plant–pathogen interactions. We finally determined the wild-type and *chs5* microbiota of distinct compartments (roots, rhizosphere, and phyllosphere) at different developmental stages. We show that the structure of the bacterial communities (as defined by Friedrich [47]) associated with wild-type or mutant backgrounds were on the whole quite similar. Most importantly, we demonstrated that bacteria affiliated to the *Streptomyces* genus colonized the wild-type and the *chs5* mutant differently, suggesting the role of one or several plant metabolites in the selection of *Streptomyces*. 

## 2. Results

### 2.1. The chs5 Mutant Exhibits Changes in Carotenoids, Xanthophylls, Phenylpropanoids, and Lipid Profiles 

The wild-type *A. thaliana* (ecotype Columbia-0) and the *chs5* mutant were grown in potting soil, in holoxenic standard temperature (STC) or low-temperature conditions (LTC). In LTC, *chs5* rosettes exhibited yellowish marginal leaf sectors and were slightly smaller than wild-type rosettes (Figure 1a, right, Appendix A). This growth phenotype is reminiscent of previous observations of an albino temperature-sensitive phenotype of *chs5* plants linked to reduced chlorophyll and carotenoid amounts [43,44]. In STC, *chs5* rosettes were visually almost identical to wild-type ones except for some tiny sectors of leaf albinism observed on a few leaves (Figure 1a, left, close-up from bottom left, red arrows indicate leaf albinism sectors, Appendix A). We measured the chlorophyll and carotenoid amounts from plants grown in STC and showed that *chs5* leaves had a very slightly reduced total amount of chlorophyll contents as compared to the wild-type leaves (Figure 1b). In addition, the number of leaves per rosette was unchanged between the wild type and *chs5* (Figure 1c). Altogether, these phenotypic traits indicate that *chs5* plants are characterized by a closely similar morphological phenotype when grown in STC. 

To better characterize this *chs5* mutant, chemical profiles were established using untargeted metabolomics. Liquid chromatography–high-resolution mass spectrometry (LC–HRMS) analysis was performed on leaf extracts. The method used enabled a large-scale metabolite annotation with confidence level 2 (42 metabolites) and level 3 (1160 metabolites) with respect to an accepted standard for HRMS-based elucidation of compounds [48]. Annotations of *m*/*z* data sets and statistical analysis of fold-change between intensities in extracts from *chs5* and wild-type genotypes grown in LTC or STC were carefully performed (Appendix A). Relevant functional metabolic segments showing variations between genotypes were defined using a chemical structural similarity enrichment analysis coined ChemRich [49] (Figure 2). This approach unveiled three major components that are accurately portraying *chs5* metabolic profile changes. The first one is a clear-cut decrease in carotenoids and to a certain extent in xanthophylls, a series of oxidized derivatives of carotenoids, in both STC (in agreement with data from Figure 1b) and LTC (in agreement with initial observations [43,44,50]). Notably, the perturbation of the terpenoid pathway in *chs5* leaves when grown in STC also influences the pool of sesquiterpenes and norisoprenoids that exhibits indisputable variations (Figure 2). The second one is a striking unbalance of phenylpropanoids (flavonoids, cinnamates, and coumarins) in STC or LTC and variations detected in groups of metabolites possibly linked to phenylpropanoids, particularly amino acids and phenols (as building blocks or precursors). This change is particularly obvious in LTC (Figure 2). The third one is a very strong change in the lipid profile of the *chs5* leaves that exhibits an increase in phosphatidylcholines (and to a lesser extent, saturated fatty acids, hydroxy fatty acids, and epoxy fatty acids) in LTC, whereas phosphatidylcholines and unsaturated fatty acids decreased in STC (Figure 2). The abundance of 36 metabolites decreased or increased in *chs5* as compared to the wild type, in both conditions (Appendix A, *p*-value < 0.05). For example, we observed in both conditions that the abundance of trigonelline, methylnicotinate, and homarine increased in the *chs5* compared to the wild type. On the other hand, we observed in the *chs5* mutant and in both conditions, a decrease in the abundance of one monoterpenoid (sporothriolide, 2,2,4,4,-Tetramethyl-6-(1-oxopropyl)-1,3,5-cyclohexanetrione) and one sesquiterpenoid (Tanacetol A).

Altogether, these data revealed that *chs5* rosettes have a defect in various metabolites. We observed changes in lipid composition depending on the growth conditions, and in phenylpropanoid composition (flavonoids, cinnamates, and coumarins) in both STC and LTC conditions. Flavonoids and coumarins are known to be involved in plant–bacteria interactions [35,51,52,53]. Consequently, the *chs5* mutant is an interesting genotype to analyze the effect of metabolic remodeling on the capacity of *A. thaliana* to interact with its microbiota or with a model bacterial pathogen, *Pseudomonas syringae* pv. *tomato* DC3000 (Pto DC3000).

### 2.2. Plant–Bacteria Interactions Are Changed in the chs5 Mutant

#### 2.2.1. The *chs5* Mutant Is More Sensitive Than the Wild Type to the Phytopathogen DC3000 Infection

We first implemented a bioassay to challenge the capacity of the *chs5* mutant to cope with one specific bacterium known to interact with *Arabidopsis*, the hemibiotrophic bacterial pathogen Pto DC3000. We considered plants grown in STC to perform the assay, because STC allows a closely similar growth of wild-type and *chs5* rosettes, preventing any bias due to morphological differences (vide supra). Such rosettes had about 15 leaves and a diameter of about 7 cm on average (Figure 1c). Disease symptoms were observed on both wild-type and *chs5*-infected plants (Appendix A). It has been previously demonstrated that the amount of stigmasterol increased upon Pto DC3000 infection and could be considered as a biochemical marker of infection by pathogens [39]. We observed a five- to sevenfold increase in stigmasterol in wild-type (*p*-value = 0.032) and, with a lower significance, in *chs5* plants (*p*-value = 0.054) after infection by Pto DC3000 (Appendix A). A significant increase in the colonization by Pto DC3000 was measured in the *chs5* leaves compared to the wild type, in three independent experiments (Appendix A). These results revealed that Pto DC3000 was able to colonize the *chs5* mutant more efficiently than the wild type, in holoxenic conditions.

#### 2.2.2. The Global Structures of the Wild-Type and *chs5* Microbiota Are Similar When Grown in STC or in LTC

The observed differential colonization of wild-type and *chs5* plants by the pathogen Pto DC3000 was indicative of a possible change in the plant–bacteria interactions to an extent that had to be determined. Therefore, in the second step, we performed a thorough analysis of the microbiota of wild-type and *chs5* plants.

We analyzed the bacterial communities of wild-type and *chs5* plants grown in STC. Here again, the rationale of the experiment was to minimize possible alterations in plant–bacteria dynamics caused by major growth differences (i.e., comparing plants with similar size and shape, as mentioned above for the Pto DC3000 bioassay). For that, we defined two major developmental stages, namely, rosettes and siliques, respectively defined as stages H and J [54] (Figure 1c). We also performed additional microbiota comparative profiling on plants grown in LTC at the silique stage, to further mine any modification of the microbiota in those conditions.

The global structure of the microbiota of *Arabidopsis* grown in STC or LTC (Figure 3 and Appendix A) was compared to previous reports [4,8,10,11]. Bacterial communities of the wild type and *chs5* are dominated by *Proteobacteria*, *Actinobacteriota,* and *Bacteroidota* phyla at both developmental stages (rosettes and siliques) in both growth conditions. α- and δ-*Proteobacteria*, *Actinobacteria,* and *Bacteroidia* were dominant at the class level (Appendix A). *Burkholderiales*, *Xanthomonadales*, and *Rhizobiales* were dominant at the order level (Appendix A). Overall, we show that the *Arabidopsis* microbial communities exhibit changes with respect to the compartments considered (Figure 3 and Appendix A). To evaluate these changes, four different α-diversity indexes were used (described in the Materials and Methods section). We observed that when grown in STC at the rosette stage (Figure 4a) or the silique stage (Appendix A), or in LTC (Appendix A), the richness and evenness of these bacterial communities are higher in the rhizosphere and roots than in the phyllosphere. Four β-diversity indexes were also used. Multi-dimensional scaling (MDS/PCoA) and hierarchical clustering based on those β-diversity indexes revealed also that the community composition and the phylogenetic diversity in the phyllosphere were different from those in the roots or rhizosphere, when plants were grown in STC (Figure 4b and Appendix A). Contrastingly, no such clear differences were observed between compartments when plants were grown in LTC (Appendix A).

A comparison of the global structure of the microbiota of wild-type and *chs5* rosettes grown in STC was performed. According to the α-diversity indexes, we observed a similar richness and evenness for both plant genotypes, in each compartment (Figure 4a). According to the β-diversity indexes illustrated by principal coordinate analyses (CoAP = MDS), the global structures of the microbiota were nearly identical between wild-type and *chs5* plants (Figure 4b). Similarly, the global structures of the microbiota of wild-type and *chs5* plants grown in STC were very close at the silique stage (Appendix A). When grown in LTC, the α-diversity indexes revealed differences between the wild-type and *chs5* (*p*-value < 0.05) (Appendix A) whereas the β-diversity indexes were similar in the mutant and the wild-type (Appendix A). These global analyses revealed that only the richness and evenness were different in LTC but not the overall community composition and phylogenetic diversity. More interestingly, the *chs5* mutation did not affect the overall composition or richness of a bacterial community in STC at the rosette or at the silique stage.

#### 2.2.3. The Colonization of Specific Bacteria Is Reduced in the *chs5* Mutant as Compared to the Wild-Type Plant, in STC at Both Stages, and in LTC

The global analyses revealed that the *chs5* mutation did not affect the overall structure or richness of the *A. thaliana* bacterial community. Nevertheless, we noticed that the relative abundance of specific OTUs changed significantly between wild-type and mutant plants. We analyzed more carefully those OTUs further named “variable OTUs or variable microbiota”.

The relative abundance of 27, 98, and 41 OTUs found at the rosette stage in the phyllosphere, rhizosphere, and roots, respectively, was significantly (*p*-value < 0.01) different between genotypes grown in STC (Appendix A). This is therefore defining a variable component of the analyzed microbiota. We found that these OTUs were affiliated to specific taxa that were enriched in the variable component of the microbiota, in all three compartments (Appendix A). For example, when considering the phyllosphere, the fractions corresponding to *Actinobacteria* and *Patescibacteria* were larger (48.1 vs. 24.1%; 7.4 vs. 2.1%) in variable OTUs than in the total microbiota, and the proportion of *Proteobacteria* was lower (33.3 vs. 43.8%). In roots, the proportion of *Proteobacteria* and *Bacteroidota* was larger in the variable OTUs than in the total microbiota (56.1 vs. 48.1%; 14.6 vs. 9.1%), whereas, in the rhizosphere, the proportion of *Patescibacteria* and *Bdellovibrionota* was larger in the variable OTU (5.1 vs. 1.7%; 6.1 vs. 3.3%) (Appendix A). This observation points out that the colonization of specific taxa is impaired in the *chs5* mutant.

We further analyzed differences at the genus level. In the roots, *Burkholderia-Paraburkholderia* represented 1.6% of the total OTUs but 14.6% of the variable OTUs, whereas *Sphingomonas* represented 7.3% of the variable OTUs but 1.7% of the total OTUs. *Allorhizobium-Neorhizobium-Pararhizobium-Rhizobium* also contributed to a significant proportion of the variable community: we found 14.8, 9.8, and 4.1% of these genera in the phyllosphere, roots, and rhizosphere, respectively, in the variable OTUs, but only 1%, in each compartment, in the total microbial community. Several of those variable OTUs were affiliated to *Streptomyces*. Those OTUs accounted for only 2.8% and 3.1% of the total microbiota of both genotypes in the phyllosphere and the roots, respectively (Appendix A). However, these OTUs represented 22.2% and 9.8% of the *chs5*-specific variable OTUs, in the phyllosphere and the roots, respectively. More specifically all variable OTUs affiliated to *Allorhizobium-Neorhizobium-Pararhizobium-Rhizobium* were always more abundant in the *chs5* mutant than in the wild type whereas all variable OTUs affiliated to *Streptomyces* (except one in the rhizosphere) were always more abundant in the wild type than in the mutant, in all three compartments (Appendix A).

We then analyzed if the colonization efficiency of those specific taxa was also modified in *chs5* grown in STC at the silique stage (Appendix A). Among the 117 OTUs whose abundance is impaired significantly (*p*-value < 0.01) in *chs5* as compared to the wild type at the silique stage (Appendix A), 48 OTUs were more abundant, and 69 OTUs were less abundant in the *chs5* mutant as compared to the wild type. Remarkably, we also observed that seven OTUs affiliated to *Streptomyces* were more abundant in the wild type than in the mutant at this developmental stage.

The abundance of 145 OTUs increased significantly (*p*-value < 0.01) in the mutant as compared to the wild type grown on LTC (Appendix A). Among these OTUs, three OTUs affiliated to *Flavobacterium*, two affiliated to *Paenibacillus,* and two affiliated to *Caulobacter* displayed changed abundances in at least two compartments (Appendix A). The abundance of 58 OTUs was significantly higher (*p*-value < 0.01) in the wild type than in the mutant plants (Appendix A). In fact, as observed in STC, nine OTUs affiliated to *Streptomyces* were more abundant in the wild type than in *chs5* (seven OTUs in the rhizosphere (Appendix A), four OTUs in the roots (Appendix A) and two OTUs in the phyllosphere (Appendix A).

#### 2.2.4. The Variation in Abundance According to the Stage of Development Is Different for Some Specific Taxa, in the Wild-Type and the Mutant Plants

To know whether the shape of the bacterial communities varied similarly during the life span of wild-type and *chs5* plants, we compared bacterial communities of wild-type or mutant plants grown in STC, at the rosette and silique stages, respectively defined as stages H and J [54] (Figure 1c). In the wild type, according to the α-diversity indexes, richness and evenness were similar at both stages when grown in STC (Appendix A). β-diversity analysis revealed that the community composition (according to the Jaccard index) and the phylogenetic diversity (according to the Unifrac index) were slightly different at the silique and rosette stages in STC (Appendix A). The abundance of 345 OTUs varied in the wild-type plants according to stages (Appendix A). The abundance of 281 OTUs increased at the silique stage as compared to the rosette stage. These OTUs were affiliated to *Acidobacteria*, *Chloroflexi*, *Myxococcota*, *Patescibacteria*, *Planctomycetota*, *Spirochaetota,* and *Verrucomicrobiota*. The abundance of 64 OTUs affiliated to *Proteobacteria*, *Firmicutes*, *Bdellovibrionota,* and *Bacteroidota* decreased at the silique stage as compared to the rosette stage (Appendix A). In the *chs5* mutant, among the 410 OTUs whose abundance varies with the stage, 324 OTUs increased and 86 OTUs decreased in abundance, at the silique stage as compared to the rosette stage (Appendix A). Thus, variations in the composition of the microbiota with respect to the developmental stage (H or J) were clearly shown in STC between the wild-type and mutant plants. These results are in agreement with previous observations showing that some bacteria vary quantitatively between plants taken at different developmental stages [10].

We then investigated whether the variations in microbiota observed between the two stages, H and J, were the same between wild-type and mutant plants. We observed that the percentage of variable OTUs affiliated to *Actinobacteria*, *Bacteroidota*, and *Patescibacteria* whose abundance decreased at the silique stage was higher in the *chs5* mutant than in the wild type (Appendix A). In addition, the percentage of variable OTUs affiliated to *Proteobacteria* and *Planctomycetota* whose abundance increased at the silique stage is higher in the wild type than in the *chs5* mutant (Appendix A). These observations suggested that the abundance of some OTUs is differently affected by the growth or developmental stage, in the wild type and in the mutant. We analyzed in detail such OTUs, whose abundance varies in *chs5* between the two stages considered in this study, whereas no variation was observed in the wild type for those OTUs. A total of 58 OTUs were affiliated to *Proteobacteria* (34.5%), *Actinobacteria* (20.7%, 5.2% *Mycobacterium*, 3.4% *Streptomyces*, and 3.4% *Aeromicrobium*), *Bdellovibrionota* (19%, 13.8% *Bdellovibrionaceae*) and *Firmicutes* (8.6%). On the other hand, 128 OTUs affiliated to *Proteobacteria* (44.8%, and among them 5.2% *Pseudomonas*, 4.7% *Deviosaceae*, and 4.1% *Sphingmonadaceae*), *Actinobacteria* (9.3%, and among them 3.5% of *Nocardioides*), *Bacteroidota* (11.6%, 5.2% *Flavobacterium*), and *Verrumicrobiota* (5.8%) were more abundant at the silique stage than at the rosette stage, in the *chs5* mutant but not in the wild type. Interestingly, 3.4% of the OTUs that were less abundant at the silique stage than at the rosette stage in the *chs5* mutant but not in the wild type (Appendix A), were affiliated to *Streptomyces*.

#### 2.2.5. Streptomyces Species Are Impaired in Their Colonization or Growth When Associated with the *chs5* Mutant

The colonization of several OTUs affiliated to the *Streptomyces* genus was changed in the *chs5* mutant when referring to the wild type regardless of growth conditions and plant developmental stage. Remarkably, the abundance of the same six *Streptomyces*-affiliated OTUs (Cluster-825, -256, -3929, -473, -36, and -186, defined in LTC) was strongly altered in the *chs5* mutant compared with the wild type, in both STC and LTC (Appendix A). To better define the affiliation of these six OTUs, a gene capture using a hybridization approach was performed to obtain near full-length 16S rRNA sequences of the *A. thaliana* microbiome [56]. In these libraries, we searched for the sequences sharing the highest identity to the six OTU sequences identified in the initial 16S metabarcoding and affiliated to *Streptomyces* (more than 99% identity, i.e., two mismatches in the 422nt-long sequence or less) (Appendix A). A deeper phylogenetic analysis was performed with these additional sequences (Appendix A). It confirmed that the six OTUs that were less abundant in the *chs5* mutant than in the wild type in all tested conditions were closely related to *S. bryophytorum*, *S. cocklensis*, or *S. paucisporeus*. These results strongly suggest that this bacterial impairment in colonizing *chs5* plants is rather not strain-specific but would concern several *Streptomyces* species. To know if bacteria affiliated to *Streptomyces* are found in the microbiota of *A. thaliana* grown under other conditions, we compared the sequences of these *Streptomyces*-affiliated OTUs to those found in plant metagenomic datasets using the BLAST tools from the Joint Genome Institute (JGI) (https://img.jgi.doe.gov/cgi-bin/mer/main.cgi (accessed on 17 August 2020)). We found several sequences sharing 100% identity with the 19 OTUs identified in this study (with a coverage of more than 200 nt), in *Arabidopsis* metagenomic data sets but also in Populus, Miscanthus, Switchgrass, Maize, or Agave metagenomic data.

## 3. Discussion

Plant organs and tissues are the hosts of microbial communities (or microbiota) that consequently form holobiont entities. The microbiota contributes to different plant biological processes such as growth and development or resistance to abiotic and biotic stresses. How specific interactions between plant and pathogenic, commensal, or mutualistic microorganisms are mediated, and how bacteria are selected by a plant are important questions to address. Here, we have evaluated the impact of a defect in the isoprenoid and lipid metabolism of the model *Arabidopsis thaliana* on plant–bacteria interactions. We have highlighted important features of the microbiota profiles of a viable mutant previously studied for its partial defect in the production of the C_5_ isoprenoid building blocks in the plastidial compartment [43,44].

The *A. thaliana chs5* mutant considered in this study offers the considerable advantage of expressing a weak allele of a key gene implied in the biogenesis of isoprenic precursors. We have extended this analysis and have shown that the *chs5* mutant had a morphology closely similar to that of the wild type in standard temperature growth conditions (STC), but a more pronounced phenotype in low-temperature growth conditions (LTC). The reasons for this temperature-dependent growth phenotype are not fully understood yet. In this work, we have also carried out a thorough untargeted metabolomic analysis. We observed a decrease in carotenoids but also in xanthophylls of the *chs5* mutant when grown in STC. Xanthophylls are carotenoid oxidized derivatives that form a large group of C_40_ lipophilic isoprenoids synthetized from the plastidial pathway [57]. Additionally, we have refined the chemical composition of the wild type and *chs5* mutant at the vegetative growth stage. Our metabolomic data revealed that the *chs5* mutant had a broader remodeling of its metabolism than expected from the initial genetic characterization [43]. In addition to isoprenoids (sesquiterpenes, diterpenes, chlorophylls, and xanthophylls), we pointed out a marked difference in phosphatidylcholines and fatty acids of the *chs5* mutant, with an increase in LTC and a decrease in STC compared to the wild type. This mutant also exhibited a modification in its phenylpropanoid composition (flavonoids, cinnamates, and coumarins) and in its amino acid and phenol content, in all growth conditions (STC and LTC). It is worth noting that phenylpropanoids as well as carotenoids and xanthophylls are components of the plant’s antioxidant defenses, by protecting against free radicals and oxidative stress [37,57,58]. The observed decrease in phenylpropanoids, carotenoids, and xanthophylls may lead to oxidative stress of the plant but also of the interacting bacteria. Those bacteria may therefore deal with higher oxidative stress that may influence the efficiency of their interactions with their host, as previously described [59]. Moreover, carotenoids and xanthophylls are precursors of abscisic acid (ABA), a phytohormone involved in plant–bacteria interactions [60,61]. Flavonoids and coumarins are also known to be involved in plant–bacteria interactions [53,62]. Overall, these data showed that the *chs5* mutant exhibited changes in the abundance of several metabolites known to be involved in plant–bacteria interactions.

Here, we have shown that the *chs5* mutant was impaired in plant–bacteria interactions (with pathogens or with microbiota). We have performed thorough profiling of microbiota colonizing the wild-type or *chs5 A. thaliana* using the relevant 16S rRNA gene sequencing method [5,63]. The data presented in this study are in agreement with previous reports describing the phylogenetic structure of *A. thaliana* microbiota: *Proteobacteria*, *Actinobacteria,* and *Bacteroidetes* were dominant phyla at both growth stages that we also considered, and microbiota changed during the plant life span. Microbiota richness and structures in the phyllosphere were different from the rhizospheric or the root microbiota, while these two microbiotas were very similar [4,5,7,8,10,11,13,14]. The data provided in this study point out very clearly marked differences in the distribution of specific OTUs between *chs5* and the wild type. It is therefore very likely that the wild-type plant may select those specific bacteria via a mechanism that is impaired in the *chs5* mutant. Several studies suggest that plants select their microbiota via metabolites produced in the root exudates [31,32,33,34,64]. Some metabolites that are less or more abundant in the *chs5* mutant may be involved in such a selection process or required for these specific OTUs to colonize the plants. These metabolites may be flavonoids that were shown to be inducers of symbiosis [35], coumarins that shape the root microbiota [62], or isoprenoids implied in the adjustment of root microbiota in *A.thaliana* [40,65,66]. Trigonelline can be degraded by *Rhizobium* [67,68]. Interestingly, we observe that this metabolite was more abundant in the *chs5* mutant as compared to the wild type, and this mutant was better colonized by *Rhizobium* bacteria.

The different composition of the microbiota between wild-type and mutant may modify direct microbial competition or antagonism between microorganisms. In the case of the *chs5* mutant, this could result in a better colonization by pathogens [69]. We challenged the behavior of the *chs5* mutant towards *P. syringae* DC3000 attack as a functional assay of the bacteria–*Arabidopsis* interactions. We showed that *chs5* was much more colonized by the pathogen than the wild type. Infections by hemi-biotrophic pathogens such as *P. syringae* DC3000 lead to the induction of a systemic acquired resistance (SAR) [34,46,70]. Nonetheless, the plant defenses rely not only on the SAR mechanism but also on other defense mechanisms that limit the spread of the pathogen. Indeed, several metabolites interfere with the pathogen growth or virulence [71,72,73]. For example, phenylpropanoids (coumarins and flavonoids) [35,51,52] or isoprenoids [39,74,75] were associated with a modified resistance to pathogens. Such changes in the metabolome also occur in the *chs5* mutant and may explain at least partly the increased colonization by *P. syringae* DC3000 in *chs5* as compared to the wild type. Alternatively, this increased colonization may be related to a change in the plant microbiome composition since plants are protected from pathogens not only by their own immunity but also by components of the microbiota [14,69,76,77,78,79]. Pathogen attenuation by commensal bacteria may be direct, via the production of antimicrobial compounds, competition for the niche or nutriments, or bacteria–bacteria inhibition of virulence. The indirect beneficial effect of the microbiota can occur via the induction of plant defenses. As a matter of fact, bacteria belonging to *Pseudomonas*, *Burkholderia*, *Rhizobium,* or *Sphingomonas* genera protect *A. thaliana* against *P. syringae* DC3000 [77,78,79,80]. We observed that the *chs5* microbiota was different from wild-type microbiota when grown in STC, in the conditions where the pathogen colonization differences were observed between the wild-type and the mutant plants. We observed that the colonization by *Allorhizobium-Neorhizobium-Pararhizobium-Rhizobium* was increased in the *chs5* mutant compared to the wild type, although this increase was insufficient to limit Pto DC3000 colonization in the mutant. Other members of the microbiota may nevertheless have such a protective role, for example, bacteria belonging to the *Streptomyces* genus. We found several *Streptomyces* phylogenetically related to *S. paucisporeus*, *S. bryophytorum,* or *S. cocklensis* in the OTUs that were more abundant in the wild type than the *chs5* mutant. This is in line with previous indications about the presence of *Streptomyces* in that model plant species [4,10,16]. These data suggest that the interactions between those *Streptomyces* and *A. thaliana* require the production of plant metabolites whose synthesis is altered in the *chs5* mutant. This decrease in *Streptomyces’* abundance could indirectly reduce the plant protection against *P. syringae*, since these bacteria are able to produce antibiotics and can help to reduce pathogen colonization [15,16,17,18].

Altogether, our results are in agreement with the hypothesis that selected *Streptomyces* bacteria are recruited by the plant via specific signals or perception mechanisms, ending up with the establishment of specific interactions and various biological or physiological implications for the holobiont [30]. Deciphering the multiplicity of molecular interactions and mechanisms that shape a microbiota is highly challenging but of prime importance for reaching a comprehensive understanding of photosynthetic holobionts.

## 4. Materials and Methods

### 4.1. Horticultural Growth Conditions

*Arabidopsis thaliana* ecotype Colombia (Col-0) plants were cultivated in growth chambers as holoxenic organisms. All seeds were kept at −20 °C for 48 h, for vernalization before sowing. Wild-type and *chs5* mutant plants [42,43] were grown in 7-cm diameter pots in soil (LAT-Terra standard topferde, Awita). The standard temperature condition (STC) was 21 °C during the light phase (6 Lumilux tubes T5, Osram; photon fluency of 250 μmol photon.s^−1^.m^−2^ measured at the level of rosettes) and 18 °C during the dark phase of an equinoctial photoperiodic regime. The low-temperature condition (LTC) was 18 °C during the light phase and 15 °C during the dark phase.

### 4.2. Infection with Pseudomonas syringae pv. tomato DC3000 (Pto DC3000)

The phytopathogen *Pseudomonas syringae* pv. *tomato* DC3000 was grown on solid Kings’ B medium (KB, 20 g/L proteose peptone (CondaLab, Madrid, Spain), 1.5 g/L K_2_HPO_4_ (Merck, Darmstadt, Germany), 15 g/L glycerol (VWR, Rosny-sous-Bois, France), 1.5 g/L MgSO_4_·7H_2_O (Merck, Darmstadt, Germany), 15 g/L agar (Sigma-Aldrich, Saint-Louis, Missouri, USA)), supplemented with 50 µg/mL rifampicin (Sigma-Aldrich, Saint-Louis, Missouri, USA), at 28 °C for 2 days. Bacteria were transferred onto liquid KB + rifampicin medium and grown to exponential phase on a rotary shaker (180 r.p.m.) at 28 °C. The bacterial culture was centrifuged at 2500× *g* for 10 min. The bacterial pellet was washed twice and resuspended in 10 mM MgCl_2_. OD_600nm_ was adjusted to obtain 10^5^ colony-forming units per milliliter (CFU/mL). Ten leaves per holoxenic *Arabidopsis* rosettes (6–8 week-old, Figure 1) were infiltrated using a needle-less syringe [46], with either 10^5^ CFU (1 mL) of *P. syringae* DC3000 or mock-infiltrated with a sterile solution of 10 mM MgCl_2_ (Merck, Darmstadt, Germany). Plants were thereafter cultivated in independent compartments of the same growth chamber to avoid contact. To enumerate the number of bacteria infecting leaves, infiltrated leaves from 8 to 12 plants from each genotype (that were distributed in at least two independent compartments) and 2 mock-treated plants were harvested at 6 days post-infection (dpi). For each plant, 10 leaf discs (3.14 cm^2^ each) were crushed in KB + rifampicin. After serial dilutions, samples were plated onto KB + rifampicin agar plates and incubated at 28 °C for 2 days and numerated.

### 4.3. Metabolomic Analysis

#### 4.3.1. Extraction and Quantification of Chlorophylls, Carotenoids, and Phytosterols

Seedlings from holoxenic rosette leaves (bulks of three to five plants described in Figure 1) were weighed and crushed in liquid nitrogen and then extracted in freshly prepared 80% acetone (Sigma-Aldrich, Saint-Louis, Missouri, USA) in water (*v*/*v*). Samples were kept in the dark at −20 °C for 24 h. Supernatants (200 μL) were transferred in 96-well microplates (PS, U-bottom, MICROLON^®^, Greiner Bio-one, Frickenhausen, Germany). Optical density was measured for each well (in triplicate) at 470 nm, 646 nm, and 663 nm on a FLUOstar Omega spectrometer (BMG Labtech^®^, Ortenberg, Germany). The concentration of chlorophylls and carotenoids in samples was calculated with established formulas [81].
c_a_ (μg·mL^−1^) = 12.25 A_663_ − 2.79 A_646_
c_b_ (μg·mL^−1^) = 21.50 A_646_ − 5.10 A_663_
c_(x+c)_ (μg·mL^−1^) = (1000 A_470_ − 1.82 c_a_ − 85.02 c_b_)/198
c_a_: concentration of chlorophyll a; c_b_: concentration of chlorophyll b; c_(x+c)_: concentration of xanthophylls and carotenes. The concentrations of chlorophylls and carotenoids were expressed in μg.mg^−1^ of the plant sample.

To measure the total amount of sterols in the acetone extracts and the corresponding pellets, these fractions were dried for 1 h at 65 °C then freeze-dried. A saponification was performed in 6% KOH in methanol (3 mL; Carlo Erba^®^, Emmendingen, Germany) at 70 °C for 2 h. After addition of 0.5 vol. of water, the unsaponifiable was extracted three times with n-hexane (Carl Roth^®^, Karlruhe, Germany). Dried residues were then incubated in an acetylation reaction mixture and steryl acetates were analyzed by GC-MS as described [82].

#### 4.3.2. Non-Targeted Metabolomic Analysis

Leaf material (ten biological replicates) collected from holoxenic plants grown in LTC or STC (Appendix A) was ground in liquid nitrogen using a Tissue Lyser apparatus (Qiagen, Hilden, Germany). Sample preparation and analysis was adapted from Villette et al. [83]. Fresh leaf powders (300 mg) were suspended in 1.5 mL cold (−20 °C) methanol spiked with an internal standard of deuterium labelled [2H6](+)-cis,trans-abscisic acid (2H6-ABA, 0.2 µg/mL). After 10 s of vortexing, the samples were kept for 16 h at −20 °C, then centrifuged at 13,000 rpm for 15 min at 4 °C. The supernatant was collected and dried by sublimation using a SpeedVac concentrator (Savant SPD121P, Thermo Fisher, Waltham, Massachusetts, USA). A second and a third extraction were performed by adding 1 mL of cold (−20 °C) methanol each time to the pellets, vortexing, and centrifugation. A fourth extraction was applied with 500 µL of ethyl acetate, vortexing, and centrifugation. All extracts were combined in the same vial and dried using a speed vacuum concentrator. Dried extracts were resuspended in 300 µL of ethyl acetate and analyzed by liquid chromatography coupled to high-resolution mass spectrometry (LC–HRMS) using an UltiMate 3000 system (Thermo, Waltham, Massachusetts, USA) coupled to an Impact II (Bruker, Bremen, Germany) quadrupole time-of-flight (Q-TOF) spectrometer.

Chromatographic separation was performed on an Acquity UPLC ^®^ BEH C18 column (2.1 × 100 mm, 1.7 µm, Waters, Milford, MA, USA) equipped with and Acquity UPLC ^®^ BEH C18 pre-column (2.1 × 5 mm, 1.7 µm, Waters, Milford, Massachusetts, USA) using a gradient of solvents A (Water, 0.1% formic acid) and B (MeOH, 0.1% formic acid). Chromatography was carried out at 35 °C with a flux of 0.3 mL/min, starting with 5% B for 2 min, reaching 100% B at 10 min, holding 100% for 3 min, and coming back to the initial condition of 5% B in 2 min, for a total run time of 15 min. Samples were kept at 4 °C, 5 µL were injected in full loop mode with a washing step after sample injection with 150 µL of wash solution (H_2_O/MeOH, 90/10, *v*/*v*). The spectrometer was equipped with an electrospray ionization (ESI) source and operated in positive and negative ion modes on a mass range from 20 to 1000 Da with a spectra rate of 8 Hz in Auto MS/MS fragmentation mode. The end plate offset was set at 500 V, the capillary voltage at 2500 V (positive mode) or 4500 V (negative mode), the nebulizer at 2 Bar, the dry gas at 8 L/min, and the dry temperature at 200 °C. The transfer time was set at 20–70 µs (positive mode) and 40.8–143 µs (negative mode) and the MS/MS collision energy was at 80–120% with a timing of 50–50% for both parameters. The MS/MS cycle time was set to 3 s, the absolute threshold to 816 cts, and active exclusion was used with an exclusion threshold at 3 spectra, release after 1 min, and the precursor ion was reconsidered if the ratio current intensity/previous intensity was higher than 5. A calibration segment was included at the beginning of the runs allowing the injection of a calibration solution from 0.05 to 0.25 min. The calibration solution used was a fresh mix of 50 mL isopropanol/water (50/50, *v*/*v*), 500 µL NaOH 1M, 75 µL acetic acid, and 25 µL formic acid. The spectrometer was calibrated on the [M^+^H]^+^/[M^−^H]^−^ form of reference ions (57 masses from *m/z* 22.9892 to *m*/*z* 990.9196 in positive mode; 49 masses from *m*/*z* 44.9971 to *m*/*z* 996.8221 in negative mode) in high-precision calibration (HPC) mode with a standard deviation below 1 ppm before the injections for each polarity mode, and re-calibration of each raw data was performed after injection using the calibration segment.

Raw data were processed with MetaboScape version 4.0 software (Bruker, Bremen, Germany). Molecular features were considered and grouped into buckets containing one or several adducts and isotopes from the detected ions with their retention time and MS/MS information when available. The parameters used for bucketing are a minimum intensity threshold of 10,000 (positive mode) or 1000 (negative mode), a minimum peak length of 3 spectra, a signal-to-noise ratio (S/N) of 3, and a correlation coefficient threshold set at 0.8. The [M^+^H]^+^, [M^+^Na]^+^, [M^+^K]^+^, and [M^+^NH4]^+^ ions (positive mode); [M^-^H]^-^ and [M^+^Cl]^−^ ion (negative mode) were authorized as possible primary and seed ions. Replicate samples were grouped and only the features found in 80% of the samples of one group were extracted from the raw data. The obtained lists of features from ion positive and negative ion modes were merged. The parameters used for metabolite annotation were as follows: The maximum allowed variation on the mass (Δ*m*/*z*) was set to 3 ppm, and the maximum mSigma value (assessing the good fitting of isotopic patterns) was set to 30. The merged list of features was annotated using SmartFormula to generate a raw formula based on the exact mass of the primary ions and the isotopic pattern. Analyte lists were derived from KNApSAcK (http://www.knapsackfamily.com/KNApSAcK_Family/ (accessed on 6 April 2020)), PlantCyc (https://plantcyc.org/ (accessed on 10 March 2020)), FooDB (http://foodb.ca (accessed on 17 March 2018)), LipidMaps (https://www.lipidmaps.org/ (accessed on 17 March 2018)), and SwissLipids (https://www.swisslipids.org/ (accessed on 11 April 2018)) to obtain a level three annotation according to Schymanski (tentative candidates based on exact mass and isotopic profile) [48]. Spectral libraries (Bruker MetaboBASE Personal Library 3.0, MoNA_LCMSMS_spectra, MSDIAL_LipidBDs-VS34) were searched to obtain level two annotations (probable structure based on library spectrum match (MS^2^ data) according to Schymanski [48].

### 4.4. Microbiota Profiling

*Arabidopsis* whole plants (6- or 8-week-old, two independent experiments) grown either in STC (see Figure 1c for their size) or LTC were gently removed from the soil. The phyllosphere, rhizosphere, and roots were separated (Appendix A). The rhizosphere (i.e., adherent soil) was scrapped from roots with a sterile scraper, and roots were shortly washed in sterile distilled water to remove as much soil as possible. The inner and surface root or leaf tissues were not discriminated and therefore designated as “root” or “phyllosphere”, respectively. Samples (1 (in LTC) or five plants (in STC) per replicates, three replicates per condition) were crushed with a pestle in a −20 °C frozen mortar. DNA from the rhizosphere (250 mg adherent soil), and from the finely ground phyllosphere and roots (50 mg) was extracted using the PowerSoil DNA Isolation Kit, and the PlantDNA Isolation kit (MO BIO Laboratories, Inc., Carlsbad, CA, USA), respectively. The DNA concentration and quality were estimated by measuring the OD at 260 nm and 280 nm. The 16S rRNA-encoding gene was PCR-amplified from 12.5 ng template DNA (Appendix A, Appendix A). Libraries were constructed according to the 16S Metagenomic Sequencing Library Preparation protocol (Illumina Part version # 15,044,223 Rev. B, available on https://support.illumina.com/documents/documentation/chemistry_documentation/16s/16s-metagenomic-library-prep-guide-15044223-b.pdf) (Appendix A) with the following modifications: 16S rRNA-encoding genes were amplified in duplicate from the extracted DNA using the primer listed in Appendix A that targets the bacterial/archaeal 16S rRNA gene variable region 5–6. The primers (Appendix A, Appendix A) used for this first PCR1 were composed of (from 5′ to 3′ ends): (1) the Illumina overhang sequence (for hybridization of sequencing primers for Read 1 and Read 2) as described in the Illumina 16S protocol; (2) a 0 to 7 pb heterogeneity spacer to increase the nucleotide diversity for sequencing, as described in Fadrosh et al. [84]; (3) the 16S V5–V6 gene-specific sequence. This first amplification (PCR1, 25 μL) was performed by mixing 25 ng genomic DNA with the KAPA HiFi HotStart ReadyMix PCR Kit (12.5 μL) (Kapabiosystems, Boston, MA, USA) and ad hoc primers (5 μL at 1 μM) using the following thermocycling conditions: initial denaturation at 95 °C for 3 min; 25 cycles of 95 °C for 30 s, 55 °C for 30 s and 72 °C for 30 s; final elongation at 72 °C for 5 min. 16S rRNA sequences were amplified (PCR1) twice independently for each sample and both amplification products were pooled and checked on 1% agarose gel for successful amplification. Those products were purified using AMPure XP beads (Agencourt, Beckman-Coulter, Brea, Californa, USA), and controlled with the DNA 1000 kit on a Bioanalyzer instrument (Agilent^®^, Agilent Technologie, Santa Clara, CA, USA) (Appendix A). The second amplification (PCR2) was carried out using the primers from the Nextera XT index kit (Illumina^®^, San Diego, CA, USA). Amplicons were purified using AMPure XP beads (Agencourt, Beckman-Coulter, Brea, CA, USA) and controlled for their size (Bioanalyzer, Agilent^®^, Agilent Technologie, Santa Clara, CA, USA). These resulting libraries were normalized and then pooled together for 2 × 300 paired-end sequencing in the presence of 5% PhiX v3 (Illumina^®^, San Diego, CA, USA, MiSeq platform).

Bioinformatic processing was performed using the FROGS version 3.1 pipeline (sourced from Migale Bioinformatics Facility, Jouy-en-Josas, France) [85] (Appendix A). Briefly, it included a pre-processing of the sequencing read data with “VSEARCH” (to delete PCR duplicates, and too-long or too-short reads). Quality sequences were clustered to operational taxonomic units (OTUs, >97% sequence similarity) with “Swarm”. Chimeric out sequences were removed using “VSEARCH”. Filtering was performed to keep OTUs present in at least three samples with minimal coverage of five sequences and suppress contaminants (phiX). Taxonomic assignments were determined using multi-affiliation output with the Silva_138_16S database. Operational taxonomic units (OTUs) classified as mitochondrial, or cyanobacteria/chloroplasts sequences were removed. The accuracy of the results was assessed by drawing rarefaction curves, to visualize the gain of diversity while adding sequences (Appendix A). Data and diversity were analyzed with the Phyloseq package [55] version 1.30.0 (sourced from Migale Bioinformatics Facility, Jouy-en-Josas, France) (Appendix A). Four different α-diversity indexes were used to study the global richness and diversity in the different compartments: the richness was evaluated using the total number of OTUs (“observed”) or the Chao1 index expressing the number of observed OTUs + estimate of the number of unobserved OTUs. The evenness (representing the phylogenetic diversity) was expressed with the Shannon index (evenness of the species abundance distribution) and the inverse Simpson index (inverse probability that two sequences sampled at random come from the same species index). Four β-diversity indexes were used to compare the different samples. The indexes called Bray–Curtis (fraction of the community specific to condition 1 or to 2) and Jaccard (fraction of species-specific to either condition 1 or 2) allowed a comparison of the composition of the communities, whereas the Unifrac (fraction of the phylogenetic tree specific to either 1 or 2) or weighted Unifrac (fraction of the diversity specific to condition 1 or to 2) allowed a comparison of the phylogenetic diversity [55]. Ordination was visualized using multidimensional scaling (MDS/PCoA) and hierarchical clustering based on these indexes.

16S rRNA sequences were submitted to GenBank (BioProject ID PRJNA577227).

### 4.5. Gene Capture

DNA extracted as described above from the rhizosphere of wild-type plants grown in LTC (100 ng) was used for the libraries’ construction with the Nextera Flex Library kit (Illumina, San Diego, California, USA). Eight PCR cycles were performed, and the resulting libraries were controlled on the Agilent Bioanalyzer (Agilent Technologies, Santa Clara, CA, USA) using a High Sensitivity DNA chip. Probe preparation and hybridization capture were performed according to published methods [56,86,87,88]. A minimal set of 15 degenerated probes targeting all known 16S rRNA genes and potential unknown sequences [56] was used to investigate bacterial and archaeal diversity. A set of 17 degenerated probes (31 to 50 bp) targeting the 18S rRNA gene was used to explore eukaryotic diversity as described in [89]. Briefly, probes were determined using the KASpOD [86] and PhylArray algorithms [90], and a custom curated database derived from all 16S or 18S rDNA sequences from the EMBL, similarly built as for PhylOPDb development [87]. Adaptor sequences were added to the ends of the probes to enable their amplification by PCR, resulting in “ATCGCACCAGCGTGT-NX-CACTGCGGCTCCTCA” sequences, with NX representing the 16S or 18S rRNA gene-specific capture probes. Biotinylated RNA capture probes were then synthesized as described by [88]. In brief, adaptors containing the T7 promoter were added to the 16S or 18S rRNA gene-specific capture probes via ligation-mediated PCR, and the final biotinylated RNA probes were obtained after in vitro transcription and purification. Hybridization capture was carried out independently for each sample and each target (16S or 18S rRNA gene) as previously described [88]. Between 200 and 300 ng of the Illumina library were used and mixed with 1 to 1.5 μg of salmon sperm DNA (Invitrogen, Carlsbad, CA, USA), denatured for 5 min at 95 °C, and incubated for 5 min at 65 °C before adding 16 μL of prewarmed (65 °C) hybridization buffer 2x (10x SSPE, 10x Denhardt’s solution, 10 mM EDTA and 0.2% SDS) and 200 to 300 ng of prewarmed (65 °C) biotinylated RNA probes. After hybridization at 65 °C for 24 h, the probe/target heteroduplexes were captured using 200 to 300 ng of washed streptavidin-coated paramagnetic beads (Dynabeads M-280 Streptavidin, Invitrogen, Carlsbad, CA, USA). The beads were collected using a magnetic stand (Ambion, Carlsbad, CA, USA) and washed once at room temperature with 500 μL 1x SSC/0.1% SDS and three times at 65 °C with 500 μL prewarmed 0.1x SSC/0.1% SDS. The captured fragments were eluted with 50 μL 0.1 M NaOH, neutralized with 70 μL of 1 M Tris–HCl pH 7.5 after supernatant collection, and PCR-amplified using primers complementary to the library adapters. To increase the enrichment efficiency, a second round of hybridization capture was performed using the first-round capture products starting from 500 ng of amplified DNA library, and using 500 ng of probes and Dynabeads. Enriched DNA samples were quantified with the Qubit Fluorometer (Invitrogen, Carlsbad, CA, USA) and checked on a Bioanalyzer 2100 (Agilent Technologies, Santa Clara, CA, USA) before sequencing on an Illumina Miseq system (2 × 250 paired-end reads).

Raw reads were trimmed for Illumina adapters using Trimmomatic (v0.38; [91]) then quality-filtered with PRINSEQ-lite PERL script v0.20.4 (min_qual_mean = 25, trim_qual_window = 3, trim_qual_step = 1, min_len = 60; [92]). The database SILVA 138 SSURef NR99 [93] was used as a reference database for subsequent analysis. Trimmed reads corresponding to rDNA were extracted using SortMeRNA (v2.1; [94]) with default parameters. Near-full length 16S and 18S rDNA sequences were reconstructed from rDNA reads using EMIRGE software and the emirge_amplicon.py script (v0.60; [95]). The parameters used were join_threshold set to 1 and 120 iterations. Taxonomical affiliation was performed using the plugin “feature-classifier sklearn classifier” from QIIME2 (v2019.1; [96]) and the full-length SILVA 138 database (provided by QIIME2), with the p-confidence set to ‘0.7′. Ambiguous sequences were further processed using the SINA aligner from the Silva website [97]. Sequencing data were submitted to ENA (PRJEB46988, ERR6668032). Only the 16S rRNA sequences were used in this study.

### 4.6. Phylogenetic Analysis

A phylogenetic analysis of the OTU sequences obtained in this study was performed using the “A la carte” mode available on the phylogeny platform (http://www.phylogeny.fr/ (accessed on 6 October 2020)), using the MUSCLE alignment (full mode), Gblocks curation, and ProtDist/FastDist+BioNJ distances options (Number of Bootstraps: 500) [98,99,100,101,102]. Similar sequences of these OTUs were downloaded by searching in the public database at NCBI using the BLAST tool (https://blast-ncbi-nlm-nih-gov.insb.bib.cnrs.fr/Blast.cgi (accessed on 6 October 2020).

### 4.7. DNA Sequence Comparisons

16S rRNA-encoding sequences sharing similarities with the OTUs characterized in this study and to *S. cocklensis* and *S. bryophytorum* were searched for in plant metagenomic data. For this, the BLAST tool (default parameters) of the “Integrated Microbial Genomes and Microbiomes” (IMG) platform of the “Joint Genome Institute” (JGI) was used. OTU sequences were compared to a subset of metagenomic sequences available on this platform and selected as “Host-associated: Plants” (381 microbiomes, accessed on 7 April 2020). The BLAST tool (megablast, default parameters) from NCBI (https://blast.ncbi.nlm.nih.gov (accessed on 12 August 2020)) was used to compare OTUs sequences obtained with metabarcoding or gene capture approaches, with sequences present in nucleotide collection (nr/nt) (GenBank+EMBL+DDBJ+PDB+RefSeq sequences, but excluding EST, STS, GSS, WGS, TSA, patent sequences as well as phase 0, 1, and 2 HTGS sequences and sequences longer than 100 Mb.).

### 4.8. Statistical Methods

#### 4.8.1. Non-Targeted Metabolomics

Statistical analyses were performed in MetaboScape 4.0 software (Bruker, Bremen, Germany) with 10 samples per group using the areas of the peaks as the unit of reference. To comply with the small number of samples, a Wilcoxon rank-sum test was used to compare the groups to each other. Compounds were considered as statistically differential between two groups using the thresholds of *p*-value ≤ 0.05 and fold change ≥2 or ≤−2. The *p*-values and fold changes from the tests were used for the chemical enrichment analysis in ChemRICH [49], keeping the differential and non-differential compounds to allow the enrichment analysis (Figure 2).

#### 4.8.2. Infection Experiments, Analysis of Stigmasterol, and Chlorophyll Accumulation

The normality and equality of variances (homoscedasticity) were verified with a Shapiro–Wilk test and a Bartlett test, respectively, and a statistical *t*-test or one-way ANOVA were used to analyze the colonization by Pto DC3000 (CFU/cm^2^) or the chlorophylls concentrations. For analysis of the stigmasterol accumulation (Appendix A), the normality and equality of variances (homoscedasticity) were estimated with a Shapiro–Wilk test and a Bartlett test, respectively. Because at least one of both conditions was not observed with our sample, significant differences were evaluated using a non-parametric Kruskal–Wallis test, and the *p*-values were evaluated with a Dunn test and the Bonferroni correction.

#### 4.8.3. Microbiota Profiling

A statistical test (ANOVA) was used to analyze if the alpha diversity index were significantly different. For the differential abundance of OTUs, the DEseq2 package incremented in R (https://www.r-project.org/, last update on 29 April 2020) was used with a Wald test to compare groups.

#### 4.8.4. Growth of the Wild Type or the *chs5* Mutant in Holoxenic Conditions

To compare the number of leaves, the diameter of the rosettes, and the length of primary stems (Figure 1c), the normality and equality of variances (homoscedasticity) were estimated with a Shapiro–Wilk test and a Bartlett test, respectively. Because at least one of both conditions was not observed with our sample, significant differences were evaluated using a non-parametric Kruskal–Wallis test, and the *p*-values were evaluated with a Dunn test and the Bonferroni correction.

## Figures and Tables

**Figure 1 ijms-23-12952-f001:**
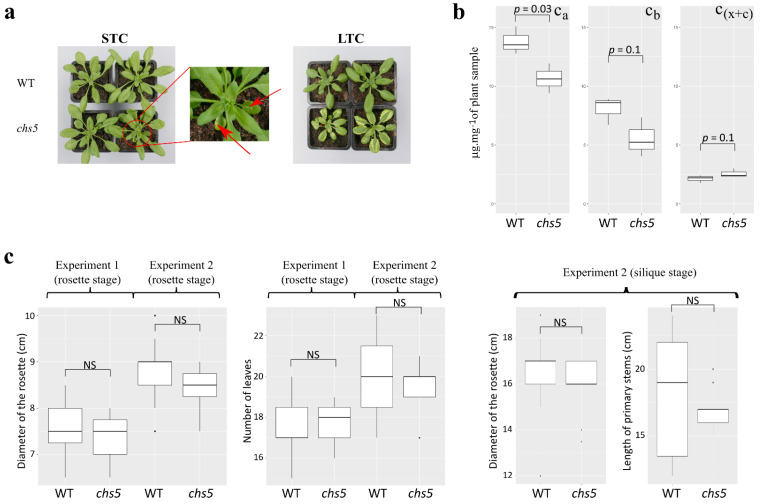
*A. thaliana* wild-type ecotype Columbia-0 (WT) and *chs5* mutant plants used in this study. (**a**) Plants were photographed at the fully expanded rosette stage, on day 42. STC, standard temperature condition; LTC, low-temperature condition. Leaves of *chs5* rosettes exhibit albino sectors in LTC whereas, in STC, *chs5* rosettes were closely resembling wild-type except for a very few tiny albino sectors shown in the close-up. (**b**) Chlorophyll content (μg.mg^−1^ plant sample) in wild-type and *chs5* plants grown in STC shown in (**a**), c_a_: concentration of chlorophyll a; c_b_: concentration of chlorophyll b; c_(x+c)_: concentration of xanthophylls and carotenoids. The normality and equality of variances (homoscedasticity) were verified with a Shapiro–Wilk test and a Bartlett test, respectively, and a statistical *t*-test was performed. Details are given in the Materials and Methods section. (**c**) Morphometrics of plants grown in STC used in this study. Two experiments were performed to provide plant material for microbiota profiling. Data for both experiments (named “Experiment 1” and “Experiment 2”) are given. The normality and equality of variances (homoscedasticity) were verified with a Shapiro–Wilk test and a Bartlett test, respectively, and a statistical one-way ANOVA test was performed. NS: No significance.

**Figure 2 ijms-23-12952-f002:**
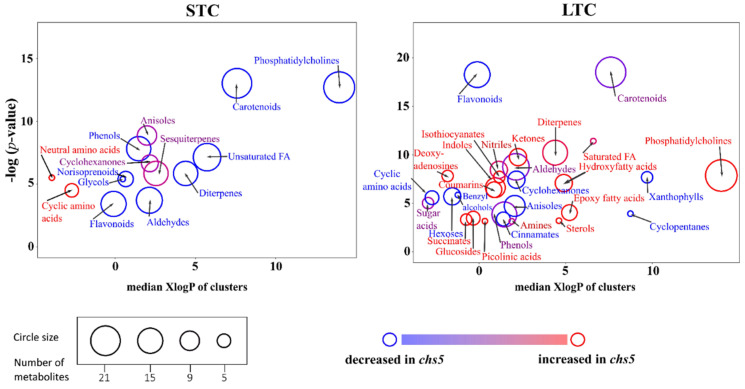
Chemical similarity enrichment analysis (ChemRICH) of metabolic profiles. Each circle represents a significantly altered family of metabolites (cluster). These clusters are defined based on chemical similarities highlighted by a hierarchical Tanimoto map (not shown). The plot *y*-axis shows the statistical relevance of the altered clusters: the *p*-value is calculated by the Kolmogorov–Smirnov test. Only enrichment clusters are shown that are significantly different at *p*-value < 0.05. The plot *x*-axis shows the increasing hydrophobicity (or decreasing hydrophilicity), from left to right along the abscissa (expressed as XlogP value). The circle color scale gives the proportion of increased or decreased compounds in each cluster: red = increased in *chs5*, blue = decreased in *chs5*, and purple-color circles have both increased and decreased metabolites. The circle sizes represent the total number of metabolites belonging to the cluster.

**Figure 3 ijms-23-12952-f003:**
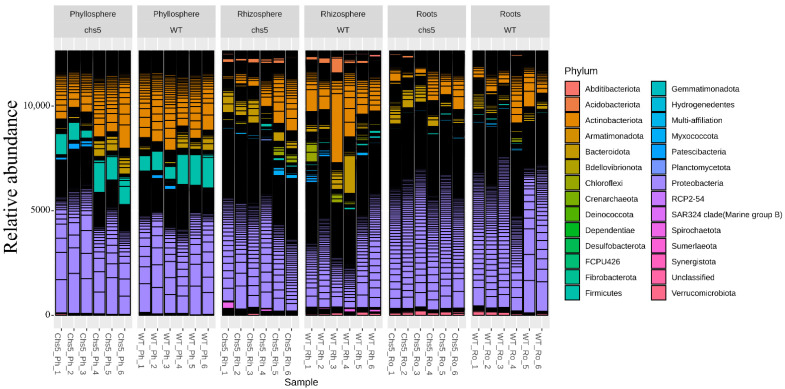
Relative abundance (%) of the different phyla identified in the *Arabidopsis* wild type (WT) and *chs5* mutant grown in STC at the rosette stage. Two biological replicates including three technical replicates were considered in each case. For each sample represented individually on the horizontal axis (x), the abundance values are represented on the vertical axis (y). The abundance values for each OTU are stacked in order from largest to smallest, separated by a thin horizontal line. The black areas represent the low abundance OTUs stacked on top of each other.

**Figure 4 ijms-23-12952-f004:**
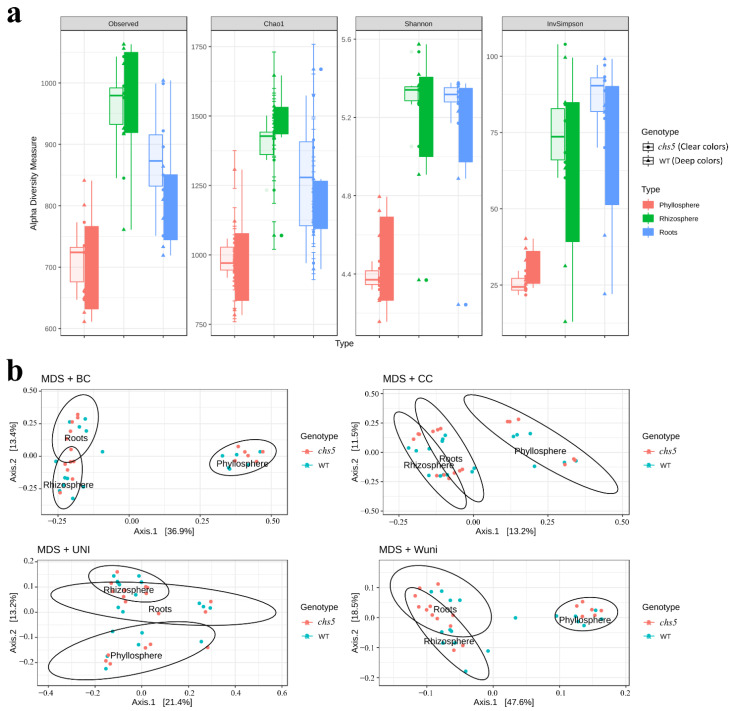
α- and β-diversity analysis of bacterial community structures of *Arabidopsis* wild type (WT) and *chs5* mutant grown in STC at the rosette stage. Two biological replicates including three technical replicates were considered in each case. (**a**) α-diversity of wild-type and *chs5* microbiota in the phyllosphere, rhizosphere, and roots. Four different α-diversity indexes were used to study the global richness and diversity (see the Materials and Methods section): the richness was evaluated using the “observed” or the Chao1 indexes; the evenness (representing the phylogenetic diversity)54was expressed with the Shannon and the inverse Simpson indexes. When grown in STC at the rosette stage, the richness and evenness are higher in the rhizosphere and roots than in the phyllosphere (ANOVA, *p*-value < 0.001). No significant difference was observed between WT and *chs5* (ANOVA, *p*-value = 0.439, 0.962, 0.278, 0.234 for the observed, Chao1, Shannon, and InvSimpson indexes, respectively). Clear colors: *chs5*; deep color: wild type. (**b**) β-diversity analysis illustrated by multidimensional scaling (MDS/PCoA). Four β-diversity indexes were used to compare the different samples (see Materials and Methods). The Bray–Curtis and the Jaccard indexes allowed us to compare the composition of the communities, whereas the Unifrac or weighted Unifrac allowed us to compare the phylogenetic diversity [55]. MDS/PCoA based on these indexes revealed that, when grown in STC, the community composition and the phylogenetic diversity were distinct in the three plant compartments. The ellipses were drawn at the 95% confidence interval of standard error and the mean value of the groups.

## Data Availability

The datasets presented in this study can be found in online repositories. The names of the repository/repositories and accession number(s) can be found in the article.

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
