# Peer review of "The Arabidopsis thaliana–Streptomyces Interaction Is Controlled by the Metabolic Status of the Holobiont"

_ijms, 2022, doi:10.3390/ijms232112952_

Round 1
Reviewer 1 Report
The article 'The Arabidopsis thaliana-Streptomyces interaction is controlled by the metabolic status of the holobiont' is interesting and well presented. However, introduction could be improved by focusing on Arabidopsis-Streptomyces interaction. The relationship between the metabolomic profiles of the wild type and mutant A. thaliana have been extensively compared with relation to the microbial interaction.
Generally I recommend the article for publication upon few minor corrections: Attached in the file with comments

Author Response
We thank the reviewer for a critical analysis of our work and recognition of the interest of our data to readers of IJMS. We have answered all the points mentioned in the document "peer-review-22670655.v1.pdf" (see attached pdf document with our answers).
Concerning the comment on the introduction we added a paragraph on Arabidopsis-Streptomyces interactions as suggested:
“Actinobacteria are ubiquitous in the plant microbiota and among this phylum, Streptomyces is the most abundant taxon [4,8,10,11]. Several bacteria affiliated to the Streptomyces genus have been recently demonstrated to play a role in Plant-Growth promoting (PGP) and biocontrol, in part because these bacteria are able to produce antibiotics [15–18]. In A. thaliana but also in Aconitum carmichaelii, bacteria belonging to the Streptomyces genus protect against biotic stresses [19,20]. This protection is linked to their ability to produce antimicrobial compounds [17,21]. Streptomyces were shown to display antagonisms against fungal or bacterial phytopathogens [22–25], or were found in disease-suppressive soils [26]. A prominent example is that of Streptomyces sp. EN27, which induces defense pathways in A. thaliana [27], whereas the rhizospheric Streptomyces sp. MR14 possesses both antifungal and PGPR capacities in tomato plants [28].”

Reviewer 2 Report
This work submitted by Graindorge with colleagues present a very interesting research of the effects of the isoprenoid mutation in Arabiodopsis on microbiota and finding a specific link between chs5 mutation and proliferation of Streptomyces.
The work is, in general, well written, but there are several typos and word spacings that needed to be addressed throughout the entire manuscript. In addition, there are a few minor points that should be considered in a revised version of the manuscript.
Result section.
Line 118: you mentioned confidence level 2 or 3. This criterion is very useful for those who studied metabolomics but very strange for the general authors. I recommend to add the explanation about confidence level in the Materials and Methods at the end of section “4.3.2”.
Discussion section.
Line 406-408: The message is confusing, would you please rephrase it?
Line 413-417: The sentence is too long and complex, I recommend you to separate it from line 416 “, these latter being -”
You mentioned in the result section that some classes of metabolite showed to be increased in chs5 mutant. However, I cannot find any comment regarding this. Is there no correlation between increased metabolites in Arabidopsis chs5 mutant and increased colonization of DC3000? I would recommend to mention it.
Materials and Methods
Line 517: You prepared the sample freshly but you expressed here as dry weight. Please use the correct unit.
Line 537-547: The sentences were repeated. Please amend it.
Figure 1 Legend.
Line 102: would you add the day of sample collection by adding “at day X”
Figure 4 Legend
The legend is continued to the next page. Why don’t you place the figure and legend on the same page?
Typos and word spacing
Line 46: slow -> slows
Line 84: chs5-> italic to plain
Line 139: <0.05 spacing
Line 156: cinnamates, coumarins -> , and coumarins
Line 164: chs5 -> italic
Line 175: spacing_ p-value= -> p-value =
Line 181: chs5 -> italic
Line 248: (P-value<0.05) spacing
Line 235: chs5 -> italic
Line 270: and -> plain
Line 290: (P-value<0.01) spacing
Line 290: ch5 -> typo
Line 295: spacing in p value
Line 315: . -> , comma to period
Line 321: STC in -> STC between (?)
Line 317 and 327: there is no coherence when you listed more than 2 things. For example, line 317 “Bdellovibrionota and Bacteroidota” whereas line 327 “Bacteroidota, and Pastescibacteria” Here, one has a comma before "and" while the other doesn’t.
Line 344: Streptomyces -> italic
Line 347: what does “this” mean?
Line 374: micro-organisms -> microorganisms
Line 421: delete period.
Line 508: spacing between “200ul”
Line 520: 2h spacing
Line 552: AutoMS/MS -> spacing
Line 553: 4 500 v -> 4,500 v
Line 573-574: the + should be superscript
Line 747: chs5 -> italic
Line 774-781: =cluster -> spacing
Line 781: value<0.01 -> spacing
References
You should double check the scientific name in the title in this section. Please convert them into italic. For example, “Arabidopsis thaliana” in lines 824 and 826 …..Pseudomonas syringae in line 873 and many.
In addition, the resolution of figures should be improved. Images and graphs are fine but the words are not readable due to low resolution. Would you please replace them in high resolution images.
Author Response
Answer to reviewer 2 comments (see also pdf file)
This work submitted by Graindorge with colleagues present a very interesting research of the effects of the isoprenoid mutation in Arabiodopsis on microbiota and finding a specific link between chs5 mutation and proliferation of Streptomyces.
We thank the reviewer for a critical analysis of our work and recognition of the interest of our data to readers of IJMS. We have answered all the points mentioned by the reviewers (see below for the details)
The work is, in general, well written, but there are several typos and word spacings that needed to be addressed throughout the entire manuscript. In addition, there are a few minor points that should be considered in a revised version of the manuscript.
Result section.
Line 118: you mentioned confidence level 2 or 3. This criterion is very useful for those who studied metabolomics but very strange for the general authors. I recommend to add the explanation about confidence level in the Materials and Methods at the end of section “4.3.2”.
We added the level definition in the material and methods as follows:
Analyte lists were derived from KNApSAcK (http://www.knapsackfamily.com/KNApSAcK_Family/), PlantCyc (https://plantcyc.org/), FooDB (http://foodb.ca), LipidMaps (https://www.lipidmaps.org/) and SwissLipids (https://www.swisslipids.org/) to obtain a level 3 annotation according to Schymanski (tentative candidates based on exact mass and isotopic profile) [48]. Spectral libraries (Bruker MetaboBASE Personal Library 3.0, MoNA_LCMSMS_spectra, MSDIAL_LipidBDs-VS34) were searched to obtain level 2 annotations (probable structure based on library spectrum match (MS² data) according to Schymanski [48].
Discussion section.
Line 406-408: The message is confusing, would you please rephrase it?
We changed this sentence as follows:
Overall, these data showed that the chs5 mutant exhibited changes in the abundance of several metabolites known to be involved in plant-bacteria interactions.
Line 413-417: The sentence is too long and complex, I recommend you to separate it from line 416 “, these latter being -”
We changed this sentence as follows:
Proteobacteria, Actinobacteria and Bacteroidetes were dominant phyla at both growth stages that we also considered and microbiota changed during the plant life span. Microbiota richness and structures in the phyllosphere were different from the rhizospheric or the root microbiota, these two microbiota were very similar.
You mentioned in the result section that some classes of metabolite showed to be increased in chs5 mutant. However, I cannot find any comment regarding this. Is there no correlation between increased metabolites in Arabidopsis chs5 mutant and increased colonization of DC3000? I would recommend to mention it.
We did not observe any correlation between increased metabolites in Arabidopsis chs5 mutant and increased colonization of DC3000. Nevertheless, we add in the results some examples of metabolites for which we observed an increase:
“For example, we observed in both conditions that the abundance of trigonelline, methylnicotinate, and homarine increased in the chs5 compared to the wild type.”
Moreover, we comment this observation in the discussion as follows:
“Trigonelline can be degraded by Rhizobium [1,2]. Interestingly, we observe that this metabolite is more abundant in the chs5 mutant as compared to the wild type, and this mutant is better colonized by Rhizobium bacteria.”
Materials and Methods
Line 517: You prepared the sample freshly but you expressed here as dry weight. Please use the correct unit.
This was changed
Line 537-547: The sentences were repeated. Please amend it.
One sentence was deleted
Figure 1 Legend.
Line 102: would you add the day of sample collection by adding “at day X”
This was added.
Figure 4 Legend
The legend is continued to the next page. Why don’t you place the figure and legend on the same page?
This was changed.
Typos and word spacing: All these suggestions were done.
Line 46: slow -> slows
Line 84: chs5-> italic to plain
Line 139: <0.05 spacing
Line 156: cinnamates, coumarins -> , and coumarins
Line 164: chs5 -> italic
Line 175: spacing_ p-value= -> p-value =
Line 181: chs5 -> italic
Line 248: (P-value<0.05) spacing
Line 235: chs5 -> italic
Line 270: and -> plain
Line 290: (P-value<0.01) spacing
Line 290: ch5 -> typo
Line 295: spacing in p value
Line 315: . -> , comma to period
Line 321: STC in -> STC between (?)
Line 317 and 327: there is no coherence when you listed more than 2 things. For example, line 317 “Bdellovibrionota and Bacteroidota” whereas line 327 “Bacteroidota, and Pastescibacteria” Here, one has a comma before "and" while the other doesn’t.
We removed the coma line 327.
Line 344: Streptomyces -> italic
Line 347: what does “this” mean?
We removed “and this”
Line 374: micro-organisms -> microorganisms
Line 421: delete period.
Line 508: spacing between “200ul”
Line 520: 2h spacing
Line 552: AutoMS/MS -> spacing
Line 553: 4 500 v -> 4,500 v
Line 573-574: the + should be superscript
Line 747: chs5 -> italic
Line 774-781: =cluster -> spacing
Line 781: value<0.01 -> spacing
References
You should double check the scientific name in the title in this section. Please convert them into italic. For example, “Arabidopsis thaliana” in lines 824 and 826 …..Pseudomonas syringae in line 873 and many.
This was checked and corrected when required
In addition, the resolution of figures should be improved. Images and graphs are fine but the words are not readable due to low resolution. Would you please replace them in high resolution images.
This was modified. However, when we include them in the word document, the resolution is not as good. We have also submitted the figures in tiff format, separately.
